# Vitamin D and COVID-19 susceptibility and severity in the COVID-19 Host Genetics Initiative: A Mendelian randomization study

**Guillaume Butler-Laporte**[1,2☯], **Tomoko Nakanishi**[1,3,4,5☯], **Vincent Mooser**[3,6], **David R. Morrison**[1], **Tala Abdullah**[1], **Olumide Adeleye**[1], **Noor Mamlouk**[1], **Nofar Kimchi**[1,7], **Zaman Afrasiabi**[1], **Nardin Rezk**[1], **Annarita Giliberti**[8], **Alessandra Renieri**[8,9], **Yiheng Chen**[1], **Sirui Zhou**[1,2], **Vincenzo Forgetta**[1], **J. Brent Richards**[1,2,3,10]*

1 Lady Davis Institute, Jewish General Hospital, McGill University, Montréal, Québec, Canada,
2 Department of Epidemiology, Biostatistics and Occupational Health, McGill University, Montréal, Québec, Canada, 3 Department of Human Genetics, McGill University, Montréal, Québec, Canada, 4 Kyoto–McGill International Collaborative Program in Genomic Medicine, Graduate School of Medicine, Kyoto University, Kyoto, Japan, 5 Japan Society for the Promotion of Science, Tokyo, Japan, 6 Canada Excellence Research Chair in Genomic Medicine, McGill University, Montréal, Québec, Canada, 7 Ruth and Bruce Rappaport Faculty of Medicine, Technion–Israel Institute of Technology, Haifa, Israel, 8 Medical Genetics, University of Siena, Siena, Italy, 9 Genetica Medica, Azienda Ospedaliera Universitaria Senese, Siena, Italy, 10 Department of Twin Research, King's College London, London, United Kingdom

☯ These authors contributed equally to this work.
* brent.richards@mcgill.ca

## Abstract

### Background

Increased vitamin D levels, as reflected by 25-hydroxy vitamin D (25OHD) measurements, have been proposed to protect against COVID-19 based on in vitro, observational, and ecological studies. However, vitamin D levels are associated with many confounding variables, and thus associations described to date may not be causal. Vitamin D Mendelian randomization (MR) studies have provided results that are concordant with large-scale vitamin D randomized trials. Here, we used 2-sample MR to assess evidence supporting a causal effect of circulating 25OHD levels on COVID-19 susceptibility and severity.

### Methods and findings

Genetic variants strongly associated with 25OHD levels in a genome-wide association study (GWAS) of 443,734 participants of European ancestry (including 401,460 from the UK Biobank) were used as instrumental variables. GWASs of COVID-19 susceptibility, hospitalization, and severe disease from the COVID-19 Host Genetics Initiative were used as outcome GWASs. These included up to 14,134 individuals with COVID-19, and up to 1,284,876 without COVID-19, from up to 11 countries. SARS-CoV-2 positivity was determined by laboratory testing or medical chart review. Population controls without COVID-19 were also included in the control groups for all outcomes, including hospitalization and severe disease. Analyses were restricted to individuals of European descent when possible. Using inverse-weighted MR, genetically increased 25OHD levels by 1 standard deviation on

**Data Availability Statement:** Covid-19 outcome GWAS summary statistics are freely available for download directly through the Covid-19 HGI

website (https://www.covid19hg.org/results/). The October 20th data freeze (v4) summary statistics were used for our study.

**Funding:** The Richards research group is supported by the Canadian Institutes of Health Research (CIHR: 365825; 409511), the Lady Davis Institute of the Jewish General Hospital, the Canadian Foundation for Innovation, the NIH Foundation, Cancer Research UK, Genome Québec, the Public Health Agency of Canada and the Fonds de Recherche Québec Santé (FRQS). GBL is supported by the CIHR, and a joint scholarship from the FRQS and Québec's Ministry of Health and Social Services. TN is supported by Research Fellowships of Japan Society for the Promotion of Science (JSPS) for Young Scientists and JSPS Overseas Challenge Program for Young Researchers. JBR is supported by a FRQS Clinical Research Scholarship. Support from Calcul Québec and Compute Canada is acknowledged. TwinsUK is funded by the Welcome Trust, Medical Research Council, European Union, the National Institute for Health Research (NIHR)-funded BioResource, Clinical Research Facility and Biomedical Research Centre based at Guy's and St Thomas' NHS Foundation Trust in partnership with King's College London. VM is supported by the Canada Excellence Research Chair Program. The funders had no role in study design, data collection and analysis, decision to publish, or preparation of the manuscript.

**Competing interests:** I have read the journal's policy and the authors of this manuscript have the following competing interests: JBR has served as an advisor to GlaxoSmithKline and Deerfield Capital. These companies had no role in the design, implementation, or interpretation of this study. No other authors have competing interests.

**Abbreviations:** 25OHD, 25-hydroxy vitamin D; COVID-19 HGI, COVID-19 Host Genetics Initiative; GWAS, genome-wide association study; IVW, inverse-variance weighted; MR, Mendelian randomization; OR, odds ratio; RCT, randomized controlled trial; UKB, UK Biobank.

the logarithmic scale had no significant association with COVID-19 susceptibility (odds ratio [OR] = 0.95; 95% CI 0.84, 1.08; $p = 0.44$), hospitalization (OR = 1.09; 95% CI: 0.89, 1.33; $p = 0.41$), and severe disease (OR = 0.97; 95% CI: 0.77, 1.22; $p = 0.77$). We used an additional 6 meta-analytic methods, as well as conducting sensitivity analyses after removal of variants at risk of horizontal pleiotropy, and obtained similar results. These results may be limited by weak instrument bias in some analyses. Further, our results do not apply to individuals with vitamin D deficiency.

## Conclusions

In this 2-sample MR study, we did not observe evidence to support an association between 25OHD levels and COVID-19 susceptibility, severity, or hospitalization. Hence, vitamin D supplementation as a means of protecting against worsened COVID-19 outcomes is not supported by genetic evidence. Other therapeutic or preventative avenues should be given higher priority for COVID-19 randomized controlled trials.

## Author summary

### Why was this study done?

- Vitamin D levels have been associated with COVID-19 outcomes in multiple observational studies, though confounders are likely to bias these associations.

- By using genetic instruments that limit such confounding, Mendelian randomization studies have consistently obtained results concordant with vitamin D supplementation randomized trials. This provides a rationale to undertake vitamin D Mendelian randomization studies for COVID-19 outcomes.

### What did the researchers do and find?

- We used the genetic variants obtained from the largest consortium of COVID-19 cases and controls, and the largest study on genetic determinants of vitamin D levels.

- We used Mendelian randomization to estimate the effect of increased vitamin D on COVID-19 outcomes, while limiting confounding.

- In multiple analyses, our results consistently showed no evidence for an association between genetically predicted vitamin D level and COVID-19 susceptibility, hospitalization, or severe disease.

### What do these findings mean?

- Using Mendelian randomization to reduce confounding that has traditionally biased vitamin D observational studies, we did not find evidence that vitamin D supplementation in the general population would improve COVID-19 outcomes.

- These findings, together with recent randomized controlled trial data, suggest that other therapies should be prioritized for COVID-19 trials.

## Introduction

SARS-CoV-2 infection has killed millions of individuals and has led to the largest economic contraction since the Great Depression [1]. Therefore, therapies are required to treat severe COVID-19 and to prevent its complications. Therapeutic development, in turn, requires well-validated drug targets to lessen COVID-19 severity.

Recently, vitamin D status, as reflected by 25-hydroxy vitamin D (25OHD) level has been identified as a potentially actionable drug target in the prevention and treatment of COVID-19 [2]. As the pre-hormone to the biologically active form calcitriol, 25OHD has been epidemiologically linked to many health outcomes [3,4]. Given calcitriol's recognized in vitro immuno-modulatory role [5], as well as observational and ecological studies associating measured 25OHD blood levels with COVID-19 [6,7], the vitamin D pathway might be a biologically plausible target in COVID-19. This could be of public health importance, given that the prevalence of vitamin D insufficiency is high in most countries, and that more than 37% of elderly adults in the US take vitamin D supplements [8]. Further, 25OHD supplementation is inexpensive and reasonably safe—thus providing a potential avenue to lessen the burden of the SARS-CoV-2 pandemic.

However, observational studies on 25OHD are prone to confounding and reverse causation bias. Confounding happens when the relationship between the exposure (25OHD) and the outcome (COVID-19) is influenced by unobserved or improperly controlled common causes. Reverse causation happens when the outcome itself is a cause of the exposure. Likewise, conclusions drawn from in vitro may not be applicable in vivo. Accordingly, randomized controlled trials (RCTs) on 25OHD supplementation have been undertaken to test its effect on disease outcomes where observational studies have supported a role for 25OHD level. However, across endocrinology, respirology, cardiology, and other specialties, these trials have most often not demonstrated statistically significant benefits [9–11]. Some RCTs have even shown a detriment to 25OHD supplementation [12]. In the field of infectious diseases, an individual patient data meta-analysis of RCTs of 25OHD supplementation [13] showed some benefit to prevent respiratory tract infections (odds ratio [OR] = 0.80; 95% CI: 0.69, 0.93). However, this effect was driven by generally benign upper respiratory tract infections, was not observed in lower respiratory tract disease (OR = 0.96; 95% CI: 0.83, 1.10), and even showed numerically worse all-cause mortality (OR = 1.39; 95% CI: 0.85, 2.27). Likewise, a recent trial on sepsis obtained a numerically higher mortality rate in patients who received 25OHD supplementation [14]. At present, we are aware of 2 RCTs testing the role of vitamin D supplementation on COVID-19 outcomes, both using high-dose vitamin D given at time of hospital admission for COVID-19. The first RCT [15] was a small trial ($n$ = 75) showing fewer intensive care unit admissions in the vitamin-D-treated arm. However, the follow-up time for mortality varied, and the open-label design put the study at high risk of bias. The second RCT [16] was a larger study ($n$ = 240) using a double-blind design, and showed no effect on mortality, risk of mechanical ventilation, and length of stay. Nevertheless, questions remain on the use of pre-illness vitamin D supplementation and its effect on disease susceptibility. While RCTs can control for confounding and provide unbiased estimates of the effect of 25OHD supplementation in COVID-19, large well-designed RCTs require considerable resources and time.

Mendelian randomization (MR) is a genetic epidemiology method that uses genetic variants as instrumental variables to infer the causal effect of an exposure (in this case, 25OHD level) on an outcome (in this case, COVID-19 susceptibility and severity) [17]. MR overcomes confounding bias since genetic alleles are randomized to the individual at conception, thereby breaking associations with most confounders. Similarly, since genetic alleles are always assigned prior to disease onset, they are not influenced by reverse causation. MR has been used in conjunction with proteomics and metabolomics to prioritize drug development and repurposing, and support investment in RCTs that have a higher probability of success [18,19]. In the case of vitamin D, MR has provided causal effect estimates consistently in line with those obtained from RCTs [9,20–24], and supporting the use of vitamin D supplementation in preventing diseases in at-risk individuals (most notably for multiple sclerosis [25]). Hence, MR may support investments in 25OHD supplementation trials in COVID-19, if a benefit was shown. Further, since MR results can be generated rapidly, such evidence may provide interim findings while awaiting RCT results.

However, MR relies on several core assumptions [26]. First, genetic variants must be associated with the exposure of interest. Second, they should not affect the outcome except through effects on the exposure (i.e., they should exhibit a lack of horizontal pleiotropy). Specifically, MR also assumes that the relationship between the exposure and the outcome is linear. However, this assumption is robust to non-liear effects as it will still provide a valid test of the null hypothesis when studying population-level effects [27], as MR then measures the population-averaged effect on the outcome of a shift in the distribution of the exposure. Third, genetic variants should not associate with the confounders of the exposure–outcome relationship. Of these assumptions, the most problematic is the second assumption. Yet, in the case of 25OHD, many of its genetic determinants reside at loci that harbor genes whose roles in 25OHD production, metabolism, and transport are well known [25]. Leveraging this known physiology can help to prevent the incorporation of genetic variants that could lead to horizontal pleiotropy.

Here, we used genetic determinants of serum 25OHD from a recent genome-wide association study (GWAS) and meta-analysis of 443,734 participants of European ancestry [28] in an MR study to test the relationship between increased 25OHD level and COVID-19 susceptibility and severity.

## Methods

We used a 2-sample MR approach to estimate the effect of 25OHD levels on COVID-19 susceptibility and severity. In 2-sample MR [29], the effect of genetic variants on 25OHD and on COVID-19 outcomes are estimated in separate GWASs from different populations. This allows for increased statistical power by increasing the sample size in both the exposure and outcome cohorts. This study is reported as per the Strengthening the Reporting of Observational Studies in Epidemiology (STROBE) guideline [30] (S1 STROBE Checklist).

Our study did not employ a prospective protocol. Analyses were first planned and performed in July 2020 and updated following peer-review in December 2020. Three major changes were made during the update. First, we used the most up-to-date COVID-19 Host Genetics Initiative (COVID-19 HGI) GWAS summary statistics. These were made available during the peer-review process. Second, to alleviate potential selection and collider bias, we modified the outcome phenotypes to include population controls. We also performed additional MR sensitivity analyses to check the robustness of our results. The latter 2 modifications were made at the request of peer-reviewers. Finally, minor changes to the results' interpretations were made following further peer-review in February 2021.

## Choice of 25OHD genetic instruments

To find genetic variants explaining 25OHD levels [28], we used a GWAS from our group, which is, to the best of our knowledge, the largest published GWAS of 25OHD levels. Importantly, this meta-analysis controlled for season of vitamin D measurement to obtain genetic variants significantly associated with 25OHD levels. From the list of conditionally independent variants provided, we further selected SNPs whose effect on 25OHD level was genome-wide significant ($p < 5 \times 10^{-8}$), whose minor allele frequency was more than 1%, and with linkage disequilibrium coefficients ($r^2$) of less than 5% (using the LDlink [31] tool and the European 1000 Genomes dataset, excluding Finnish populations). For SNPs that were not available in the outcome GWAS or with palindromic alleles of intermediate frequency (between 42% and 58%), we again used the LDlink [31] tool to find genetic proxies in the European 1000 Genomes dataset (excluding Finnish populations) using a linkage disequilibrium $r^2$ of 90% or more.

## COVID-19 outcome definitions and GWASs

We used the COVID-19 HGI outcome definitions and GWAS summary statistics for COVID-19 susceptibility, hospitalization, and severe disease outcomes [32]. For all outcomes, a COVID-19 infection defined as a positive SARS-CoV-2 infection (e.g., RNA RT-PCR or serloogy test), electronic health record evidence of SARS-CoV-2 infection (using International Classification of Diseases or physician notes), or self-reported infections from the patients. The susceptibility phenotype compared COVID-19 cases with controls, which were defined as any individuals without a history of COVID-19. The hospitalized outcome compared cases, defined as hospitalized patients with COVID-19, and controls, defined as any individuals not experiencing a hospitalization for COVID-19, which includes those without COVID-19. The severe disease outcome cases were defined as hospitalized individuals with COVID-19 who died or required respiratory support. Respiratory support was defined as intubation, continuous positive airway pressure (CPAP), bilevel positive airway pressure (BiPAP), continuous external negative pressure, or high-flow nasal cannula. Controls for the severe COVID-19 outcome were defined as individuals without severe COVID-19 (including those without COVID-19). The inclusion of COVID-19-negative participants as controls in each outcome decreases the possibility of collider bias [33] and allows for better population-level comparisons. These 3 outcome phenotypes are referred to as C2, B2, and A2, respectively, in the COVID-19 HGI documentation.

For our study, we used the 20 October 2020 (v4) COVID-19 HGI fixed effect meta-analysis of GWASs from up to 22 cohorts, performed in up to 11 countries. Every participating cohort was asked to provide summary statistics from a GWAS on the above 3 outcomes, and including the non-genetic covariates age, sex, age × age, and age × sex; 20 genetic principal components; and any locally relevant covariates at the discretion of participating studies (e.g., hospital, genotype panel). Cohorts were asked to follow common sample and variant quality control, and performed analysis only if they enrolled 100 cases or more. Analyses were done separately for each major ancestry group to further control for population stratification. For the purposes of our study, we used the meta-analysis results from European ancestry cohorts, except for the severe COVID-19 outcome, for which this meta-analysis was not available. Further details on the 3 phenotypes and participating cohorts are found in Table 1 and S1 Data.

## Primary MR analysis

The effect of 25OHD level on COVID-19 outcomes was obtained for each SNP by using the Wald ratio method. The effect of each SNP was given in standardized log-transformed 25OHD level. Each estimate was meta-analyzed using the inverse-variance weighted (IVW)

**Table 1. Sources of data for the analysis.**

| Phenotype | Source of genetic variants | |
|---|---|---|
| | **Cohort** | **Participants** |
| 25OHD circulating levels | Manousaki et al. [28] | Meta-analysis of 2 25OHD GWASs:<br>• 401,460 adult white British participants from the UKB<br>• 42,274 from an international consortium of adult individuals of European ancestry |
| COVID-19 susceptibility | Susceptibility | Meta-analysis of 22 GWASs performed in individuals of European ancestry from 11 countries:<br>• **Cases:** 14,134 individuals with COVID-19 by laboratory confirmation, chart review, or self-report<br>• **Controls:** 1,284,876 individuals without confirmation or history of COVID-19 |
| COVID-19 severity | Hospitalized | Meta-analysis of 13 GWASs performed in individuals of European ancestry from 11 countries:<br>• **Cases:** 6,406 hospitalized individuals with COVID-19<br>• **Controls:** 902,088 individuals without hospitalization with COVID-19 |
| | Severe disease | Meta-analysis of 12 GWASs performed in individuals of European ancestry from 9 countries:<br>• **Cases:** 4,336 SARS-CoV-2-infected hospitalized individuals who died or required respiratory support (intubation, CPAP, BiPAP, continuous external negative pressure, or high-flow nasal cannula)<br>• **Controls:** 623,902 without severe COVID-19 |

COVID-19 susceptibility and severity outcomes are taken from the COVID-19 Host Genetics Initiative. See S1 Data for details on cohorts of COVID-19 susceptibility and severity phenotypes.

25OHD, 25-hydroxy vitamin D; BiPAP, bilevel positive airway pressure; CPAP, continuous positive airway pressure; GWAS, genome-wide association study; UKB, UK Biobank.

method, and we performed variant heterogeneity tests to check the robustness of IVW results. Allele harmonization and computations were performed using the TwoSampleMR package [34].

## Horizontal pleiotropy sensitivity analysis

We undertook multiple analyses to assess the risk of horizontal pleiotropy (a violation of the second MR assumption). First, we used the MR–Egger method, which allows for an additional intercept (alpha) term, which also provides an estimate of directional horizontal pleiotropy. This method relies upon the assumption that the size of the direct effect of a genetic variant on the outcome that does not operate through the exposure is independent of the variant's effect on the exposure. Given possible instability in MR–Egger estimates [35], we also used the bootstrap MR–Egger method to meta-analyze the causal effect estimates from each SNP instrument. Further, we used 4 additional meta-analysis methods known to be more robust to the presence of horizontal pleiotropy (at the expense of statistical power): penalized weighted median, simple mode, weighted median, and weighted mode [36].

Second, we restricted our choices of SNPs to those whose closest gene is directly involved in the vitamin D pathway. These genes have an established role in vitamin D regulation through its synthesis (*DHCR7/NADSYN1* and *CYP2R1*), transportation (*GC*), and degradation (*CYP24A1*) (S1 Fig). This decreases the risk of selecting a genetic variant that affects COVID-19 outcomes independent of its effect on 25OHD levels.

Third, we used the PhenoScanner tool [37,38] on the remaining SNPs to check for variants associated (at a genome-wide significant threshold of $p < 5 \times 10^{-8}$) with phenotypes at risk of affecting COVID-19 outcomes independent of 25OHD, making them at higher risk of

horizontal or vertical pleiotropy. Note that vertical pleiotropy, which happens when the COVID-19 outcome is influenced by a phenotype directly in the causal pathway between 25OHD level and COVID-19 outcome, does not violate MR assumptions.

### Research ethics

Each cohort included in this study received its respective institutional research ethics board's approval to enroll patients. All information used for this study is publicly available as deidentified GWAS summary statistics.

## Results

### Choice of 25OHD genetic instruments

We obtained our 25OHD genetic instruments from our previously published GWAS on circulating 25OHD levels in 401,460 white British participants in the UK Biobank (UKB) [39], which was meta-analyzed with a GWAS on 25OHD levels of 42,274 participants of European ancestry [40]. Of the 138 reported conditionally independent SNPs (explaining 4.9% of the 25OHD variance), 100 had a minor allele frequency of more than 1%, of which 77 were directly available in the COVID-19 HGI GWAS summary statistic and had a linkage disequilibrium coefficient of less than 5%. Additionally, 3 more variants had good genetic proxies ($r^2 \geq 90\%$) and were therefore added to our instrument lists, for a total of 80 variants. These explained 4.0% of the variance in 25OHD serum levels. The full list of SNPs used can be found in S2 Data.

### COVID-19 outcome definitions and GWASs

Using the COVID-19 HGI results restricted to cohorts of European ancestry, we used a total of 14,134 cases and 1,284,876 controls to define COVID-19 susceptibility, 6,406 cases and 902,088 controls to define COVID-19 hospitalization, and 4,336 cases and 623,902 controls to define COVID-19 severe disease. Table 1 summarizes the definition and sample size of both the exposure and outcome GWASs. Since the UKB was used in the 2 phases of the MR study, some overlap between the exposure and the outcome GWASs was unavoidable (S1 Data).

### Primary MR analysis

We first used IVW meta-analysis to combine effect estimates from each genetic instrument. For a standard deviation increase in log-transformed 25OHD level, we observed no statistically significant effect upon odds of susceptibility (OR = 0.95; 95% CI: 0.84, 1.08; $p$ = 0.44). Of note, in the UKB, the distribution of 25OHD levels has a mean of 48.6 nmol/L and a standard deviation of 21.1 nmol/L. This standard deviation is comparable to what can be achieved with vitamin D supplementation, especially over short therapeutic courses [41]. Similarly, we observed no significant difference in risk of hospitalization (OR = 1.09; 95% CI: 0.89, 1.33; $p$ = 0.41) or risk of severe disease (OR = 0.97; 95% CI: 0.77, 1.22; $p$ = 0.77) associated with a standard deviation increase in log-transformed 25OHD level (Table 2; Fig 1).

### Horizontal pleiotropy assessment and sensitivity analysis

Using the MR–Egger intercept terms, we did not observe evidence of horizontal pleiotropy. While they have less statistical power than IVW meta-analysis, the 6 sensitivity meta-analyses we used also showed no evidence of an association between 25OHD levels and COVID-19 susceptibility, hospitalization, and severe disease, with each confidence interval crossing the null

**Table 2. Mendelian randomization results.**

| Outcome | Number of SNPs* | IVW OR (95% CI) | IVW p-value | IVW SNP heterogeneity p-value | Egger alpha | Alpha p-value |
|---|---|---|---|---|---|---|
| **25OHD primary analysis with all SNPs** | | | | | | |
| Susceptibility | 80 | 0.95 (0.84, 1.08) | 0.44 | 0.009 | 0.003 (−0.004, 0.009) | 0.39 |
| Hospitalization | 80 | 1.09 (0.89, 1.33) | 0.41 | 0.065 | 0.0004 (−0.010, 0.011) | 0.93 |
| Severe disease | 80 | 0.97 (0.77, 1.22) | 0.77 | 0.140 | 0.008 (−0.004, 0.020) | 0.17 |
| **25OHD sensitivity analysis restricted to genes in the vitamin D pathway** | | | | | | |
| Susceptibility | 11 | 0.94 (0.81, 1.08) | 0.39 | 0.204 | 0.002 (−0.024, 0.029) | 0.86 |
| Hospitalization | 11 | 1.04 (0.75, 1.46) | 0.81 | 0.003 | 0.028 (−0.033, 0.089) | 0.39 |
| Severe disease | 11 | 0.92 (0.68, 1.25) | 0.59 | 0.117 | 0.044 (−0.008, 0.096) | 0.13 |
| **25OHD sensitivity analysis after removal of SNPs identified by PhenoScanner** | | | | | | |
| Susceptibility | 9 | 0.91 (0.71, 1.17) | 0.48 | 0.110 | 0.002 (−0.034, 0.038) | 0.91 |
| Hospitalization | 9 | 1.02 (0.61, 1.73) | 0.93 | 0.008 | 0.012 (−0.065, 0.089) | 0.77 |
| Severe disease | 9 | 1.05 (0.64, 1.73) | 0.85 | 0.127 | 0.032 (−0.038, 0.103) | 0.40 |

Confidence intervals were obtained using normal approximations.

*Number of SNPs retained for this analysis.

25OHD, 25-hydroxy vitamin D; IVW, inverse-variance weighted; OR, odds ratio; SNP, single nucleotide polymorphism.

in the primary analysis using all SNPs (Fig 1; S1 Table). Our results are therefore unlikely to be strongly biased by horizontal pleiotropy.

We also restricted our analysis to SNPs that reside close to the 4 genes directly involved in 25OHD metabolism. This left 11 SNPs, explaining 2.9% of 25OHD variation. Using IVW, each standard deviation increase in log-transformed 25OHD was again not associated with COVID-19 susceptibility (OR = 0.94; 95% CI: 0.81, 1.08; $p$ = 0.39), hospitalization (OR = 1.04; 95% CI: 0.75, 1.46; $p$ = 0.81), and severe disease (OR = 0.92; 95% CI: 0.68, 1.25; $p$ = 0.59). For the 3 phenotypes, the MR–Egger intercept term did not support bias from directional horizontal pleiotropy.

Lastly, we used the PhenoScanner [37,38] tool to check if the SNPs used in the MR study were associated with other phenotypes. Using PhenoScanner, rs11723621 was associated with white blood cell level, and rs6127099 was associated with glomerular filtration rate [42,43]. In both cases, the association with each phenotype was mild compared to the SNP's effect on 25OHD level, as rs11723621 explained less than 0.03% of the variance in white blood cell count, and rs6127099 explained less than 0.001% of the glomerular filtration rate variance. Removing these SNPs from the 11 SNPs above further decreased the proportion of 25OHD variance explained to 1.4%. While confidence intervals widened, effect estimates when restricting our analysis to these SNPs remained null for susceptibility (OR = 0.91; 95% CI: 0.71, 1.17; $p$ = 0.48), hospitalization (OR = 1.02; 95% CI: 0.61, 1.73; $p$ = 0.93), and severe disease (OR = 1.05; 95% CI: 0.64, 1.73; $p$ = 0.85).

## Genetic instrument heterogeneity

Overall, our results showed little evidence of heterogeneity of effect between our genetic instruments (Table 2). We nonetheless observed that for at least 1 of the 3 analyses, we would have rejected the null hypothesis of homogeneous genetic effects in the COVID-19 hospitalization phenotype. However, given the large number of hypotheses tested, this may be due to chance.

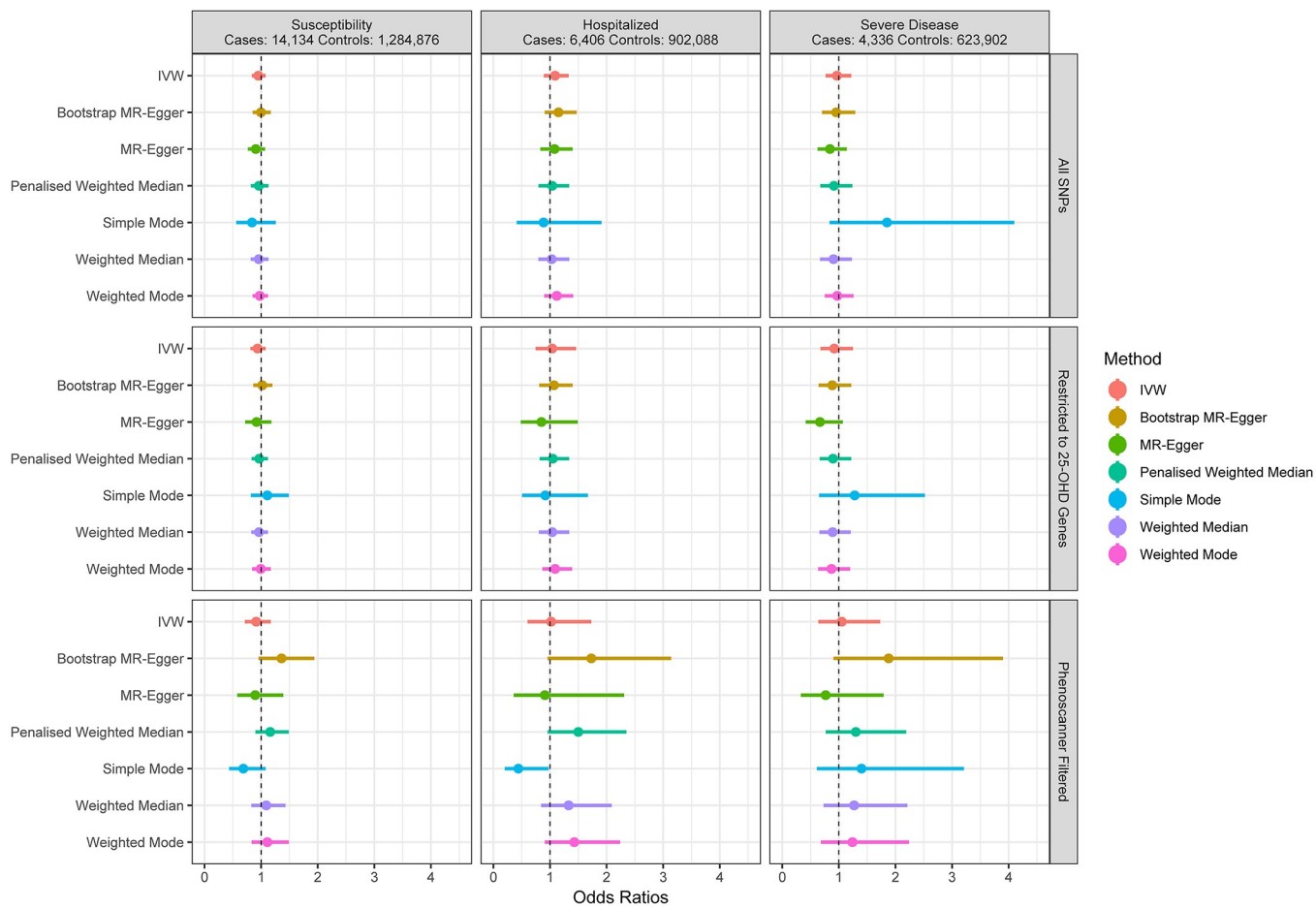

**Fig 1. Odds ratio point estimates and 95% confidence intervals for the effect of a 1-SD increase in 25OHD levels (on the log scale) on COVID-19 susceptibility and severity.** Restricted to 25-OHD Genes: analysis restricted to SNPs near the 4 genes involved in known vitamin D metabolic pathways. PhenoScanner Filtered: analysis restricted to the 4 genes above, and with removal of SNPs identified as having other associations in PhenoScanner. Full results including odds ratios, confidence intervals, and *p*-values are available in S1 Table. 25OHD, 25-hydroxy vitamin D; IVW, inverse-variance weighted; MR, Mendelian randomization.

## Discussion

In this large-scale MR study, we did not find evidence to support increasing 25OHD levels in order to protect against COVID-19 susceptibility, hospitalization, or severity. This lack of evidence was consistent across phenotypes, sensitivity analyses, and choice of genetic instruments. Differences between our findings and those reported in observational studies [6] may reflect the fact that associations between vitamin D and COVID-19 may be confounded due to factors difficult to control for even with advanced statistical adjustments, such as socioeconomic status, institutionalizaton, or medical comorbidities associated with lower vitamin D levels. While our study assessed the association between genetically determined levels of 25OHD and COVID-19, these results can still inform us on the role of vitamin D supplementation. Specifically, in contrast to observational studies, our findings do not support an association between higher 25OHD level and better COVID-19 outcome, and therefore do not support the use of vitamin D supplementation to prevent COVID-19 outcomes. Further, while one randomized trial [15] showed a benefit of vitamin D supplementation, it used an endpoint at risk of bias due to the unblinded intervention (admission to the critical care unit) and had a small sample size (*n* = 75); a larger, double-blinded randomized trial [16] of 240 patients

showed no effect of a single high dose of vitamin D3 on mortality, length of stay, or risk of mechanical ventilation. Thus, findings from the largest randomized trial to date are concordant with our MR results.

Our study's main strength is MR's track record of predicting RCT outcomes for multiple medical conditions [9–11,21–24,44,45]. Our study also leverages, to our knowledge, the largest cohort of COVID-19 cases and controls currently available (even outside of genetic studies) and the largest study on genetic determinants of 25OHD levels to date. Using these data sources, we were able to obtain results robust to multiple sensitivity analyses.

Our study still has limitations. First, our results do not apply to individuals with vitamin D deficiency, and it remains possible that truly deficient patients may benefit from supplementation for COVID-19-related protection and outcomes. However, individuals who are found to have frank vitamin D deficiency, should undergo replacement for bone protection. Second, our study may suffer from weak instrument bias, especially within sensitivity analyses that restricted to smaller sets of genetic instruments. In 2-sample MR, this bias would tend to make estimates closer to the null. Nonetheless, similar studies have been able to use MR to establish an association between 25OHD levels and other diseases (most notably multiple sclerosis [25]), suggesting that these instruments are strong enough to find such associations. Further, given the large percentage of individuals from the UKB shared between the vitamin D exposure GWASs [28] and the severe COVID-19 phenotype GWASs, this analysis is close to a 1-sample MR, which would show bias towards the observational study association. Given that this analysis also shows largely null effects, we do not suspect that weak instruments bias is a significant issue in our results. Third, given that vitamin D levels are affected by season (with higher levels after sunlight exposure), even if our SNP instruments were obtained from a GWAS that controlled for season of blood draw, effect attenuation by averaging the effect of 25OHD levels on COVID-19 over all seasons may influence results. Nevertheless, a recent study in a Finnish cohort (where sun exposure greatly varies by season) showed that genetic determinants of 25OHD level were able to discriminate between individuals with a predisposition to varying levels of 25OHD, regardless of the season [46]. Therefore, while the cyclical nature of 25OHD level is not completely modeled by MR, the size of this bias is likely small. Fourth, our MR analyses assume a linear exposure–outcome relationship. While this may slightly bias our results, simulation studies have previously shown that this assumption provides adequate results when looking at a population effect [27]. Therefore, for the purpose of vitamin D supplementation in the general population, our conclusions should still be valid. However, as pointed out above, we are not able to test the effect of vitamin D deficiency on COVID-19 outcomes. Lastly, as we only studied the effect of 25OHD and COVID-19 in individuals of European ancestry, it remains possible that 25OHD levels might have different effects on COVID-19 outcomes in other populations. However, previous RCTs on vitamin D supplementation have given similar results in populations of various ancestries [44,45].

In conclusion, using a method that has consistently replicated RCT results from vitamin D supplementation studies in large sample sizes, we find no evidence to support a protective role for higher 25OHD in COVID-19 outcomes. Specifically, vitamin D supplementation as a public health measure to improve COVID-19 outcomes is not supported by this MR study. Most importantly, our results suggest that investment in other therapeutic or preventative avenues should be prioritized for COVID-19 RCTs.

## Supporting information

**S1 STROBE Checklist. STROBE case–control study checklist.**
(DOC)

**S1 Data. Cohorts used for each outcome phenotype for the COVID-19 Host Genetics Initiative.**
(DOCX)

**S2 Data. Genetic instrument summary statistics.**
(DOCX)

**S1 Fig. Vitamin D metabolism pathway and genes involved.**
(DOCX)

**S1 Table. Results from Mendelian randomization sensitivity analyses.**
(DOCX)

**S1 Text. Acknowledgment of data contributors and the COVID-19 Host Genetics Initiative.**
(DOCX)

**S2 Text. GEN-COVID Multicenter Study.**
(DOCX)

## Acknowledgments

We thank the patients and investigators who contributed to the COVID-19 HGI (S1 Text) and the Vitamin D GWAS consortium. Members of the GEN-COVID study are acknowledged in S2 Text. This research was conducted using the UKB resource (project number: 27449).

## Author Contributions

**Conceptualization:** Guillaume Butler-Laporte, Tomoko Nakanishi, Vincent Mooser, David R. Morrison, Yiheng Chen, Sirui Zhou, Vincenzo Forgetta, J. Brent Richards.

**Data curation:** Guillaume Butler-Laporte, Tomoko Nakanishi, Annarita Giliberti, Alessandra Renieri.

**Formal analysis:** Guillaume Butler-Laporte, Tomoko Nakanishi, Sirui Zhou, Vincenzo Forgetta, J. Brent Richards.

**Funding acquisition:** J. Brent Richards.

**Investigation:** Guillaume Butler-Laporte, Tomoko Nakanishi, Vincent Mooser, David R. Morrison, Tala Abdullah, Olumide Adeleye, Noor Mamlouk, Nofar Kimchi, Zaman Afrasiabi, Nardin Rezk, Annarita Giliberti, Alessandra Renieri, Yiheng Chen, Sirui Zhou, Vincenzo Forgetta, J. Brent Richards.

**Methodology:** Guillaume Butler-Laporte, Tomoko Nakanishi, Yiheng Chen, Sirui Zhou, Vincenzo Forgetta, J. Brent Richards.

**Project administration:** Vincent Mooser, Vincenzo Forgetta, J. Brent Richards.

**Resources:** J. Brent Richards.

**Software:** Guillaume Butler-Laporte, Vincenzo Forgetta, J. Brent Richards.

**Supervision:** Vincent Mooser, Noor Mamlouk, Vincenzo Forgetta, J. Brent Richards.

**Validation:** Guillaume Butler-Laporte, Tomoko Nakanishi, David R. Morrison, Tala Abdullah, Olumide Adeleye, Nofar Kimchi, Zaman Afrasiabi, Nardin Rezk, Annarita Giliberti, Alessandra Renieri, Yiheng Chen, Sirui Zhou, Vincenzo Forgetta, J. Brent Richards.

**Visualization:** Guillaume Butler-Laporte, Tomoko Nakanishi.

**Writing – original draft:** Guillaume Butler-Laporte, Tomoko Nakanishi, J. Brent Richards.

**Writing – review & editing:** Guillaume Butler-Laporte, Tomoko Nakanishi, Vincent Mooser, David R. Morrison, Tala Abdullah, Olumide Adeleye, Noor Mamlouk, Nofar Kimchi, Zaman Afrasiabi, Nardin Rezk, Annarita Giliberti, Alessandra Renieri, Yiheng Chen, Sirui Zhou, Vincenzo Forgetta, J. Brent Richards.

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
