## [Editor Report · Decision Letter 0]

4 Sep 2020

Dear Dr Richards, 

Thank you for submitting your manuscript entitled "Vitamin D and Covid-19 Susceptibility and Severity: a Mendelian Randomization Study" for consideration by PLOS Medicine.

Your manuscript has now been evaluated by the PLOS Medicine editorial staff as well as by an academic editor with relevant expertise, and I am writing to let you know that we would like to send your submission out for external peer review.

Kind regards,

Caitlin Moyer, Ph.D.,

Associate Editor

PLOS Medicine

---

## [Decision Letter · Decision Letter 1]

23 Nov 2020

Dear Dr. Richards,

Thank you very much for submitting your manuscript "Vitamin D and Covid-19 Susceptibility and Severity: a Mendelian Randomization Study" (PMEDICINE-D-20-04227R1) for consideration at PLOS Medicine. 

Your paper was evaluated by a senior editor and discussed among all the editors here. It was also sent to three independent reviewers, including a statistical reviewer. The reviews are appended at the bottom of this email and any accompanying reviewer attachments can be seen via the link below:

[LINK]

In light of these reviews, I am afraid that we will not be able to accept the manuscript for publication in the journal in its current form, but we would like to consider a revised version that addresses the reviewers' and editors' comments. Obviously we cannot make any decision about publication until we have seen the revised manuscript and your response, and we plan to seek re-review by one or more of the reviewers. 

We expect to receive your revised manuscript by Dec 14 2020 11:59PM. Please email us (plosmedicine@plos.org) if you have any questions or concerns.

We look forward to receiving your revised manuscript. 

Sincerely,

Caitlin Moyer, PhD 

Associate Editor

PLOS Medicine

plosmedicine.org

1.Abstract: Please structure your abstract using the PLOS Medicine headings (Background, Methods and Findings, Conclusions). Please combine the Methods and Results sections into one section, “Methods and findings”.

2. Abstract: Methods: Please provide some background demographic information on the participants contributing to the GWAS used to obtain variants associated with 25OHD, GWAS for Covid-19.

3. Abstract: Line 85: Please provide the p values associated with the OR along with the 95% CIs.

4. Abstract: Line 86-87: This sentence could be clarified or expanded upon, as it seemed from the results that there was some variation in the findings between the primary and secondary/sensitivity analyses.

5. Abstract: In the last sentence of the Abstract Methods and Findings section, please describe the main limitation(s) of the study's methodology.

6. Abstract: Conclusions: Please address the study implications without overreaching what can be concluded from the data; the phrase "In this study, we observed ..." may be useful. At lines 91-92, we suggest softening the language/recommendation here-“...individuals should not use vitamin D supplements…” to focus instead on what the data do and do not support (e.g. “...these findings do not provide supporting evidence for the use of vitamin D…”)

7. Author Summary: At this stage, we ask that you include a short, non-technical Author Summary of your research to make findings accessible to a wide audience that includes both scientists and non-scientists. The Author Summary should immediately follow the Abstract in your revised manuscript. This text is subject to editorial change and should be distinct from the scientific abstract. Please see our author guidelines for more information: https://journals.plos.org/plosmedicine/s/revising-your-manuscript#loc-author-summary

8. Throughout text: Please use square brackets for in-text citations, like this [1].

9. Methods: Prospective protocol: Did your study have a prospective protocol or analysis plan? Please state this (either way) early in the Methods section.

10. Results: Lines 276-281: Please clarify for the MR Egger estimates of 25OHD and hospitalization that effects did not reach statistical significance, in contrast to what you saw with the primary IVW analysis for hospitalization.

11. Results: Line 273, 287: In these two instances, the term "trend" is used- once where statistical significance is not reached, and once where it is reached. The term “trend” should be used only when the test for trend has been conducted. Please revise accordingly.

12. Results: Line 315: Please clarify if this is the result for susceptibility with the extended phenotypes definition.

13. Results: In presenting the results, it would be helpful to more explicitly point out consistencies and inconsistencies between analyses by clarifying where secondary analyses differed from the main results, although you do allude to this where you mention widening confidence intervals (for example, at line 300-303: “While confidence intervals widened, effect estimates when restricting our analysis to these 22 SNPs remained similar for susceptibility (0.77; 95% CI: 0.48, 1.23; P=0.27), hospitalization (2.89; 95% CI: 1.18, 7.06; P=0.02), and severe disease (2.52; 95% CI: 0.63, 10.0; P=0.19), though this did not reach statistical significance in the latter case.”

14. Discussion: Please slightly re-organize the Discussion as follows: a short, clear summary of the article's findings; what the study adds to existing research and where and why the results may differ from previous research; strengths and limitations of the study; implications and next steps for research, clinical practice, and/or public policy; one-paragraph conclusion.

15. Figure 1: This diagram might be more appropriate as a supporting information file. Please define all abbreviations used in the figure in the Figure legend. It might be helpful to call attention to particular points of the pathway most relevant for your study (e.g. particular genes involved.

16. Figure 2 and supporting information Figure 2.1 (in Supplement 3): Please provide the labels for the X axis.

17. Checklist: Please ensure that the study is reported according to the STROBE guideline, and include the completed STROBE checklist as Supporting Information. When completing the checklist, please use section and paragraph numbers, rather than page numbers. Please add the following statement, or similar, to the Methods: "This study is reported as per the Strengthening the Reporting of Observational Studies in Epidemiology (STROBE) guideline (S1 Checklist)."

18. Data availability statement: Thank you for noting that the SNPs are available in published studies. If it is possible, it would be helpful to have a summary supporting information file, or a more specific reference (DOI/accession number) to find the repository genetic instrument SNPs used in your study.

Comments from the reviewers:

Reviewer #1: Overall comments: Thank you very much for the opportunity to review this manuscript entitled "Vitamin D and Covid-19 Susceptibility and Severity: a Mendelian Randomization Study". This is an interesting topic of research, which should be of importance to Covid-19. However, prior to publication, I feel that the manuscript needs some substantial work to evaluate the reliability of the results and explain to the readers how to interpret the results. 

The authors used two-sample Mendelian randomization analysis to determine the role of vitamin D status in Covid-19. As the authors suggest, MR is a powerful method for inferring causality, which seems to replace RCTs. However, its careless use can lead to many false results, especially for two-sample Mendelian randomization study. The authors made a conclusion "genetically increased 25OHD levels did not protect against Covid-19 susceptibility, or severity, and in some analyses was associated with worsened outcomes, which support vitamin D supplementation to prevent Covid-19 outcomes". Actually, the essence of MR is used to explore the causal relationship of risk factors to the outcome, but it could not replace RCT to explore the effect of reducing risk factors on the outcome. Therefore, I don't think this conclusion is appropriate. How this finding can guide the impact of vitamin D supplements on COVID-19 needs to be further explored.

 

Specific Comments:

Abstract:

1. I would reword "GWASs of Covid-19 susceptibility and severity from the Covid-19 Host Genetics Initiative were used to test the effect of 25OHD levels on these outcomes" It's not entirely clear what the authors mean. Do they mean that GWASs of Covid-19 susceptibility and severity from the Covid-19 Host Genetics Initiative were used to perform instrument-outcome associations?

2. The sentence "the genetically increased 25OHD levels tended to increase the odds ratio of severe disease, while OR = 2.21; 95% CI: 0.87-5.55" should be reworded. The finding is not consistent with its explanation.

3. Needs for more detailed information of Extensive sensitivity analyses "Extensive sensitivity analyses probing the assumptions of MR provided consistent estimates.". In addition, the result seems to be inconsistent in different instrumental variables and different methods.

Introduction

4. Needs some rewording: " In the case of vitamin D, MR has been able to provide causal effect estimates consistently in line with those obtained from RCTs, and would therefore support investments in 25OHD supplementation trials in Covid-19, if a benefit was shown". How to balance the findings of MR study and RCT?

5. Aside from the mentioned several core assumptions of MR, it is important to note the linearity assumptions within the IV framework.

Method

6. The authors should provide heterogeneity tests of individual genetic variants to explore the robust of IVW method.

7. More sensitivity analysis methods, including but not limited to the weighted median and penalized weighted median methods, should be carried out.

8. As far as I know, the COVID-19 host genetics initiative provided the summary data for severe COVID-19, COVID-19 and hospitalized COVID-19 with control groups were subjects from the general population without the specific phenotype, subjects who were COVID-19 negative based on prediction or self-report, or subjects who had COVID-19 without hospitalization. The author used extended phenotypes, but limited the controls as subjects who were COVID-19 negative based on prediction or self-report, or subjects who had COVID-19 without hospitalization. The authors should give a reason for the selected controls.

9. The definition of exposure is unclear. The detailed information of exposure- vitamin D and the information of summary data for the selected SNP associated with vitamin D should be provided.

Result 

10. The description of the results seems to be inconsistent with the results in the table, such as, the MR Egger results are inconsistent with Egger OR as the Table 2 showed. In addition, the result seems to be inconsistent in different instrumental variables and different methods.

Discussion

11. The increasing observational studies indicate that vitamin D deficiency is associated with increased COVID-19 risk[1, 2]. The authors should explain more deeply for the contradictory results in the discussion. In addition, the findings should be interpreted more cautiously for guiding future 25OHD supplementation trials.

12. However, the unmeasured confounders or alternative causal pathways may be still affected those results because of the limitation of the method. A previous study found that serum 25OHD concentrations are highly heritable is winter season, which suggests a varied role of genetic factors, dependent on the UVB intensity[3]. So how to reduce the bias of UVB against results? I suggested that it should be added as a limitation.

13. As mentioned above, it is important to note the linearity assumptions within the IV framework. If the effect of vitamin D supplementation on Covid-19 may be nonlinear, how to interpret this finding?

Referrences

1) Meltzer Do Auid- Orcid: --- Fau - Best TJ, Best Tj Fau - Zhang H, Zhang H Fau - Vokes T, Vokes T Fau - Arora V, Arora V Fau - Solway J, Solway J. Association of Vitamin D Deficiency and Treatment with COVID-19 Incidence. LID - 2020.05.08.20095893 [pii] LID - 10.1101/2020.05.08.20095893 [doi].

2) Mitchell F. Vitamin-D and COVID-19: do deficient risk a poorer outcome? (2213-8595 (Electronic)).

3) Karohl C, Su S Fau - Kumari M, Kumari M Fau - Tangpricha V, Tangpricha V Fau - Veledar E, Veledar E Fau - Vaccarino V, Vaccarino V Fau - Raggi P, et al. Heritability and seasonal variability of vitamin D concentrations in male twins. (1938-3207 (Electronic)).

Reviewer #2: In this paper authors investigate whether genetically predicted vitamin D levels are associated with Covid-19 susceptibility and severity using Mendelian randomisation approach. The rationale for the work is very well justified, and the work conducted is impressive. However, I have a few major comments that would need to be addressed before this study is published:

1. Context relating to respiratory tract infections and vitamin D RCTs is misrepresented 

Dozens of RCTs investigating the role of vitamin D in respiratory tract infections were conducted to date; majority show benefit (either statistically significant, or non-significant with OR suggestive of benefit). In particular see: 

Martineau AR et al. Vitamin D supplementation to prevent acute respiratory tract infections: systematic review and meta-analysis of individual participant data. BMJ. 2017;356:i6583. Published 2017 Feb 15. doi:10.1136/bmj.i6583

In the Introduction, statement associated with Reference 14 (line 123) therefore misrepresents the context. Of note, cited paper found increased risk in treated arm for upper respiratory tract infections, but not for lower respiratory tract infections - later probably being more relevant in the context of Covid-19.

Moreover, Castillo et al. published a RCT that high-dose vitamin D supplementation significantly reduced need for ICU (ie decreased risk of severe disease) - findings that are at odds with results reported here.

Castillo ME, Entrenas Costa LM, Vaquero Barrios JM, Alcala Dıaz JF, Miranda JL, Bouillon R, Quesada Gomez JM, E¨ ffect of Calcifediol Treatment and best Available Therapy versus best Available Therapy on Intensive Care Unit Admission and Mortality Among Patients Hospitalized for COVID-19: A Pilot Randomized Clinical study¨, Journal of Steroid Biochemistry and Molecular Biology (2020)

2. Need more information on cohorts, and on control definition and risk of misclassification 

Cohorts are referenced, but information given in the paper (or supplementary) is not enough to critically understand the work and implications, particularly in relation to cohorts used - descriptive analysis covering: age, gender, period covered (case/control assessment during which months of 2020?), countries included, ethnicity, comorbidities, BMI…). 

I suggest expanding Table 1 to include this for all cohorts used.

Among cohorts used, are disease outcomes harmonised?

Misclassification (likelihood of false negatives in particular) is very likely and it would be good to report what it might be, for example using prevalence of disease in the cohort and what was in the population cohort was sampled from at that time. Eg in UKBiobank it is reasonable to expect a high proportion of [mild or asymptomatic] cases among controls given that testing was not widespread - which also means mild cases are "missing" amongst cases. 

3. Meaning behind the reported findings

What does "one standard deviation on the logarithmic scale" of genetically increased 25OHD level mean - ie. what is the corresponding increase in nmol/L in circulating 25OHD concentration associated with this?

Given that vitD genetic proxy explains 1-4% of variance, it probably means that only a very small difference in circulating 25OHD (couple of nmol/L perhaps?) can be explained by the genetic proxy. Combined with the increasingly accepted notion that the relation between vitamin D and health outcomes follows a sigmoid (with threshold effect) rather than linear curve (Heaney R, 2013, Nutrition Reviews), is this finding strong enough to inform guidelines on vitamin d supplementation?

4. Interpretation of the findings

Looking at main results table (Table 2), it would also be true to say: 

"When we used genetic instruments that are most likely to affect Covid-19 outcomes via their effect on 25OHD (after remoal of SNPs identified by Phenoscanner), a statistically significant reduction in Covid-19 susceptibility was noted among those with higher genetically predicted 25OHD (OR=0.38, 95%CI: 0.19-0.75). A suggestive decrease in risk was also noted when analysis only included genes in the vitamin D pathway (OR=0.61, 95%CI: 0.37-1.01)."

Of note, these results are reported incorrectly in the current version of the manuscript (line 278: "….we…found.. increased 25OHD levels on susceptibility …".) - in all analysis, DECREASED risk for susceptibility was found (clear in Figure 2). 

It would be a more balanced paper if these findings were brought forward and discussed in the paper also - particularly how to explain different direction of the effect for susceptibility and severity. 

Reviewer #3: This is an interesting paper, but I'm afraid that the results are strongly affected by selection / collider bias. For me, the primary analysis is valid, but the secondary analyses are faulty. The reason is that the primary analysis is confirmed absence versus presence of COVID-19, whereas the secondary analyses are severe versus mild COVID - hence even controls are selected as those who have COVID-19. A better analysis would be severe COVID versus no COVID. As per Table 1 of "Contextualizing selection bias in Mendelian randomization: how bad is it likely to be?" by Gkatzionis and Burgess, when selection into the sample depends on the risk factor (which is clearly the case for the secondary analyses here, as COVID status of participants would associate with vitamin D levels), bias is in the opposite direction as the observational association. Hence the COVID-19 severity analyses provide associations in the harmful direction. Hence the paper is interest, but currently mainly as an exposition of the bias that can arise due to improper selection of controls.

Overall, this is an interesting paper, but I would encourage the authors to only perform analyses where the comparison group is population-based controls. My personal preference is severe COVID versus control, as severe COVID is less susceptible to selection (see https://www.medrxiv.org/content/10.1101/2020.06.18.20134676v1.full.pdf). But there are other GWAS datasets that are available (see https://www.medrxiv.org/content/10.1101/2020.09.15.20165886v1.full.pdf - although we did also consider severe versus mild disease, so not fully following my own advice!).

I think a paper on this subject would be interesting, but the results and message would likely change substantially if the authors concentrated on the primary analysis (which suggests a null result), so will only provide broad comments at this point:

1. Abstract: "Genetically increased 25OHD levels by one standard deviation on the logarithmic scale had no clear effect on susceptibility but tended to increase the odds ratio..." - I'd prefer to see non-causal language in the results section of the manuscript. What you observe is that "Genetically higher 25OHD levels... associate with the risk of hospitalization". What you infer from this is that 25OHD is not a causal risk factor, but you do not show this directly - the causal aspect of the manuscript is an inference from the findings, not the finding itself. In other places too - for me, causal language is fine when describing the aims and interpretation of the work (although should be clear that causal interpretation is subject to untestable assumptions), but not when describing the results of analyses.

2. Did you account for correlation between genetic variants? Even if the variants were conditionally independent in their associations with 25OHD, they may well be correlated in their distributions.

3. Given the recommendation to focus on the results versus controls that suggested associations in the protective direction (albeit not statistically significant), you may want to think again about weak instrument bias and the preference for one-sample MR. I don't feel strongly here, but if the results change, then the argument that weak instrument bias is unimportant as the estimates are in the harmful direction no longer holds.

4. A further limitation is that summary data MR cannot address nonlinearity. This is not a serious limitation for the main public health question, which relates to large-scale population interventions, but it may still be that vitamin D supplementation has a benefit for those who are vitamin D deficient.

In summary, if this paper is re-written to focus on the primary analysis and other analyses that suggest a null effect, then it would be a valuable contribution to the literature.

---

Stephen Burgess

[LINK]

---

## [Decision Letter · Decision Letter 2]

8 Feb 2021

Dear Dr. Richards,

Thank you very much for submitting your revised manuscript "Vitamin D and Covid-19 Susceptibility and Severity: a Mendelian Randomization Study" (PMEDICINE-D-20-04227R2) for consideration at PLOS Medicine. 

Your paper was evaluated by a senior editor and discussed among all the editors here. It was also sent to the three original reviewers, including a statistical reviewer. The reviews are appended at the bottom of this email and any accompanying reviewer attachments can be seen via the link below:

[LINK]

In light of these reviews, we would like to consider a revised version that addresses the remaining reviewers' and editors' comments. Please do address the concerns of Reviewer 2 in your response. Obviously we cannot make any decision about publication until we have seen the revised manuscript and your response, and we may seek re-review by one or more of the reviewers. 

We expect to receive your revised manuscript by Feb 15 2021 11:59PM. Please email us (plosmedicine@plos.org) if you have any questions or concerns.

We look forward to receiving your revised manuscript. 

Sincerely,

Caitlin Moyer, Ph.D.

Associate Editor 

PLOS Medicine

plosmedicine.org

Title: Please include some information on the study population if feasible.

Abstract: Please rename the “Introduction” section to “Background”

Abstract: Please spell out Mendelian randomization for MR at first use.

Abstract: Background: Line 72-73: Please revise to mitigate causal language: “Here, we used two-sample MR to assess evidence supporting a causal effect of circulating 25OHD levels on Covid-19 susceptibility.” or similar.

Abstract: Methods and Findings: Please provide some detail regarding this GWAS: “Genetic variants strongly associated with 25OHD levels in a 443,734-participant genomewide association study (GWAS) were used as instrumental variables.”

Abstract: Methods and Findings Line 79-80: Please clarify this sentence- are these two cohorts of individuals with COVID-19, or is one intended to be without COVID-19. Also, please clarify briefly the manner of how it was determined that the individuals had COVID-19.

Abstract: Conclusion: We suggest adding an opening sentence such as: “In this 2 sample MR study, we did not observe evidence to support an association between 25OHD levels and COVID-19 susceptibility, severity, or hospitalization.” or similar.

Author summary: Why was the study done?: We suggest deleting the first bullet point.

Author summary (and throughout the text): Please consistently capitalize the Mendelian in Mendelian randomization.

Methods: Line 220-221: Please qualify this statement with “to the best of our knowledge” or similar.

Methods: Line 237-243: Please clarify the description to indicate, for the hospitalization and severity cohort, whether the controls were individuals without severe Covid/hospitalized for Covid and did or did not have Covid-19.

Methods: Please add the following statement, or similar, to the Methods: "This study is reported as per the Strengthening the Reporting of Observational Studies in Epidemiology (STROBE) guideline (S1 Checklist).”

Results: Line 320: Please change “clear” to “statistically significant” or similar, and please revise the sentence to: “For a standard deviation increase in log-transformed 25OHD level, we observed no significant association with odds of susceptibility (OR = 0.97; 95% CI:0.85, 1.10; P = 0.61).”

Results: Line 321-324: To address the comment of Reviewer 2, please clarify the relationship between 1SD increase in biomarker vs instrument.

Results: Line 329: The word “clear” is ambiguous, it could be removed or replaced.

Results: Line 324-326: Please revise to “Similarly, we observed no significant difference in risk of hospitalization (OR =1.11; 95% CI: 0.91, 1.35; P = 0.30) or risk of severe disease (OR = 0.93; 95% CI: 0.73, 1.17; P = 0.53) associated with a standard deviation increase in log-transformed 25OHD level (Table 2 and Figure 1).”

Results: Line 331: To avoid overly causal language we suggest referring to associations when reporting the results. 

Results Line 338-341: Please revise to “Using IVW, each standard deviation increase in log-transformed 25OHD was again not associated with Covid-19 susceptibility (OR = 0.962; 95% CI: 0.83, 1.11;P = 0.594), hospitalization (OR = 1.07 [95% CI: 0.78, 1.47]; P = 0.668) and severe disease (OR = 0.869; 95% CI: 0.635, 1.19; P = 0.378).”

Results: Line 347-350: Please revise to “In both cases, the association with each phenotype was mild compared to their effect on 25OHD level, as rs11723621 explained less than 0.03% of the variance in white blood cell counts, and rs6127099 explained less than 0.001% of the glomerular filtration rate variance[39,40].”

Results: Line 360: Please change “hypothesis” to “hypotheses”

Discussion: Line 364-366: We suggest revising slightly to avoid overly causal language: “In this large-scale MR study, we did not find evidence to support that genetically increased 25OHD levels are protective against Covid-19 susceptibility, hospitalization, or severity. This lack of evidence supporting a causal association was consistent across phenotypes, sensitivity analyses, and choice of genetic instruments.” or similar.

Discussion: Line 371-374: Please include the citation for the study described here. Also we would suggest revising to remove the first instance of the word “small” as it is emphasized later in the statement, and also clarify the “flawed endpoint” to make your point more apparent: “Further, while a small randomized trial showed benefit of vitamin D supplementation, this trial used a flawed endpoint and a small sample size, and it is therefore unable to invalidate our results.”

Discussion: Line 404: Please revise to “...for the purpose of vitamin D supplementation…”

Discussion: Line 406: Please revise to “...as we only studied the effect of 25OHD and Covid-19 in individuals of European ancestry, it remains possible…”

Checklist: We agree the utility of the STROBE for case-control studies is limited, and the STROBE-MR does not seem to have been finalized/published. If the authors feel the STREGA extension on the STROBE items is most appropriate, please use that.

Figure 1: Would it be possible to include a note in the legend that p values are given in Supporting Information table 4 (if I have that correct)?

Page 11: The sections Data Availability, Funding Source, Competing Interest, and Transparency statement can be removed from the main text of the manuscript. Please ensure that all information is entered in the appropriate place in the manuscript submission forms (Financial Disclosure, Competing Interests, Data Availability)

Comments from the reviewers:

Reviewer #1: Authors have answered my comments. I still have a minor suggestion, as follows,

The main finding that genetically increased 25OHD levels did not protect against COVID-19 susceptibility, hospitalization, or severity, indicates that endogenous 25OHD levels are not causallly associated with COVID-19 susceptibility, hospitalization, or severity, but not completely equal to the main conclusion. I suggest the authors to make the conculusion more carefully.

Reviewer #2: 

Findings from this study have much improved after re-analysis with the most up-to-date dataset available. Process is streamlined, results clearer and presentation improved.

It is unfortunate however that other sections of the paper have deteriorated since the original submission and paper needs substantial editing for coherence, typos, accuracy and most notably context and interpretation of findings.

For example:

- abstract line 80 should presumably state "…1,284,876 WITHOUT Covid-19…"

- authors inconsistently use "vitamin D level" when they mean "genetically predicted 25OHD level" (eg Author Summary)

Major concerns at this stage are as follows:

1. In terms of methodology, the key issue with genetic instruments in this case is not that the instrument is weak, but that it's association with outcome is time-varying. 

Genetic instruments explain large fraction of the variance in winter, but virtually none in the summer. This issue has been flagged in various MR contexts eg.

https://www.ncbi.nlm.nih.gov/pmc/articles/PMC5123677/

https://academic.oup.com/aje/article/188/1/231/5098387

In the discussion section, authors need to explain how the time-varying relationship between the instrument and the biomarker might have affected their null-findings, given that this has not been incorporated in the analysis.

2. The level of biased presentation remains highly concerning. 

For example, the original submission cited selected finding from one RCT on respiratory tract infections, that showed an increased risk with vitamin D. After the individual-patient data meta-analysis of ~30 RCTs that published in BMJ was brought to authors' attention (https://www.bmj.com/content/356/bmj.i6583) - a study that showed stat sign beneficial effect of vitD overall (adjusted odds ratio 0.81, 0.72 to 0.91), and statistically significant effect among vitD deficient with substantial effect size (adjusted odds ratio 0.30, 0.17 to 0.53), authors choose to cite the statistically insignificant but "numerically worse all-cause mortality" outcome form this study (introduction line 160). 

In the refs to support the agreement between non-significant MR and RCTs, authors provide refs (Introduction line 179), but fail to mention that while statistically not significant, "numerically beneficial" effect of vit D (or OR at 1) was found in majority of primary and secondary analysis reported in these. 

Authors do not mention vitD MR studies that found a stat sign beneficial effect of genetically predicted vitD level, eg https://www.ncbi.nlm.nih.gov/pmc/articles/PMC4582411/

Authors might consider citing observational vitD study in UK Biobank 

https://www.ncbi.nlm.nih.gov/pmc/articles/PMC7204679/

3. How did authors calculate that 1 SD in genetically predicted 25OHD corresponds to 21.1 nmol/L?

Mean 25OHD in entire UK Biobank is 48.6 nmol/L and SD is 21.1 nmol/L. One SD in the instrument is not equal to 1 SD in the biomarker, particularly if instrument explains only ~4% of the variance. 

Given the authors' previous study (ref 27), it should be easy and it would be informative to use actual UK Biobank data to show the correlation between instruments (genetically predicted 25OHD) and actual 25OHD level. For example, creating quintiles of genetically predicted vitD level and giving mean and SD of measured 25OHD for each quintile in genetically predicted 25OHD.

4. Interpretation of findings is not in line with results. 

Eg last line of Abstract/results states that results do not apply to vitD deficient individuals. Appropriate conclusion should therefore be: "Our results do not support that individuals who are vitamin D sufficient be advised to take vitD supplements."

In terms of guiding future research, a more constructive recommendation should be building on well-known issues with vitamin D RCTs - for example recruiting vitamin D deficient individuals in the trial (it is established that high proportion of vitD sufficient individuals in the trial limits the study from detecting the beneficial effect, as no futher benefit can be expected among those who are vitD sufficient at baseline). Or recommendation for future MR studies in this area - eg using a design that allows for a time-varying relationship that exists between the instrument and biomarker.

In terms of using MR to prioritise drugs for testing in RCT, there are two things to consider:

1) many potential medicines do not have a naturally occurring biomarker and cannot be interrogated in MR setting so it's hard to rank them against those that do

2) vit D is suitable for co-administration and hence factorial trial design

Reviewer #3: The authors have responded appropriately to the reviewer comments, and the paper is much improved. Couple of very minor comments (in no order of importance!):

1. Line 336: "Second, we restricted out analysis" should be "Second, we restricted our analysis".

2. Line 188: "However, this assumption still provides valid results" - For me, this could be clearer. The property is this: causal estimation in Mendelian randomization assumes a linear causal relationship between the exposure and outcome. But if the relationship is truly non-linear, then the estimate is still a valid test of the causal null hypothesis, and the estimate represents the population-averaged effect of a shift in the distribution of the exposure.

3. "rejected the null hypothesis of lack of heterogeneity" - Would "homogeneity" be clearer than "lack of heterogeneity"? Also, it's not clear what is being referenced here - heterogeneity/homogeneity in what? I presume from context in the variant-specific causal estimates, but it's not clear from the text ("heterogeneity in the Covid-19 hospitalization phenotype" seems to suggest something else).

4. Line 281 - "that effects Covid-19 outcomes" -> "that affects Covid-19 outcomes"

5. Acknowledgement: the Supplements here are incorrectly numbered.

[LINK]

---

## [Editor Report · Decision Letter 3]

5 Mar 2021

Dear Dr. Richards,

Thank you very much for re-submitting your manuscript "Vitamin D and COVID-19 Susceptibility and Severity in the COVID-19 Host Genetics Initiative: a Mendelian Randomization Study" (PMEDICINE-D-20-04227R3) for review by PLOS Medicine.

I have discussed the paper with my colleagues and the academic editor. I am pleased to say that provided the remaining editorial and production issues are dealt with we are planning to accept the paper for publication in the journal.

[LINK]

We look forward to receiving the revised manuscript by Mar 12 2021 11:59PM.   

Sincerely,

Caitlin Moyer, Ph.D.

Associate Editor 

PLOS Medicine

plosmedicine.org

Requests from Editors:

1. Title: Please capitalize the first word in the subtitle, and use sentence case throughout: “Vitamin D and COVID-19 susceptibility and severity in the COVID-19 Host Genetics Initiative: A Mendelian randomization study"

2. Abstract: Line 89: Please revise "clear" to "significant" or "statistically significant" in this sentence.

3. Abstract: Conclusion: Line 100: Please change “mean” to “means” in this sentence.

4. Author Summary: What do these findings mean? We suggest revising the first bullet point, to focus more on the findings/advance of your particular study- “Vitamin D is a highly confounded variable, and traditional observational studies are at high risk of biased estimates.”

5. Author Summary: What do these findings mean? We suggest revising to: “Taken together with literature supporting concordance between MR studies and RCTs investigating vitamin D effectiveness, these findings suggest that other therapeutic and preventative avenues should be prioritized for COVID-19 trials.” or similar.

6. Throughout: Please include a space between the preceding word and bracket of the in-text citation.

7. Introduction: Line 212: Please revise the final sentence: “...to test the relationship between genetically increased 25OHD level on COVID-19 susceptibility and severity.” or similar, to avoid causal implications.

8. Methods: Line 289: We suggest “undertook multiple analyses” rather than “undertook extensive analysis” here.

9. Discussion: Lines 392-395: Please clarify this sentence: “These findings highlight the confounded association between vitamin D and COVID-19 due to factors such as older age, institutionalization, or medical comorbidities, that are all linked to lower vitamin D levels and cannot be controlled for even when using advanced statistical adjustments.” Perhaps: “Differences between our findings and those reported in observational studies [references] may reflect the fact that associations between vitamin D and COVID-19 may be confounded due to factors difficult to control for even with advanced statistical adjustments, such as older age, institutionalizaiton or medical comorbidities linked to lower vitamin D levels.” or similar.

10. Reference List: Please double check the formatting of all references, and please use the "Vancouver" style for reference formatting, and see our website for other reference guidelines https://journals.plos.org/plosmedicine/s/submission-guidelines#loc-references. 

For example, please indicate that # 16 is a preprint. For # 25, PLOS Med should be PLoS Med.

11. Table 1: Please define all abbreviations in the legend, including GWAS, UKB, CPAP, BiPAP.

12. Checklist: Thank you for including the STROBE checklist. Please make your responses more distinct from the checklist items, if possible (italics for your notes, or a separate column on the right).

[LINK]

---

## [Editor Report · Decision Letter 4]

31 Mar 2021

Dear Dr Richards, 

On behalf of my colleagues and the Academic Editor, Cosetta Minelli, I am pleased to inform you that we have agreed to publish your manuscript "Vitamin D and COVID-19 susceptibility and severity in the COVID-19 Host Genetics Initiative: A Mendelian randomization study" (PMEDICINE-D-20-04227R4) in PLOS Medicine.

PRESS

Sincerely, 

Caitlin Moyer, Ph.D. 

Associate Editor 

PLOS Medicine